# Real-World Experience of the Comparative Effectiveness and Safety of Combination Therapy with Remdesivir and Monoclonal Antibodies versus Remdesivir Alone for Patients with Mild-to-Moderate COVID-19 and Immunosuppression: A Retrospective Single-Center Study in Aichi, Japan

**DOI:** 10.3390/v15091952

**Published:** 2023-09-19

**Authors:** Jun Hirai, Nobuaki Mori, Daisuke Sakanashi, Wataru Ohashi, Yuichi Shibata, Nobuhiro Asai, Hideo Kato, Mao Hagihara, Hiroshige Mikamo

**Affiliations:** 1Department of Clinical Infectious Diseases, Aichi Medical University Hospital, 1-1 Yazakokarimata, Nagakute-shi 480-1195, Aichi, Japan; hirai.jun.326@mail.aichi-med-u.ac.jp (J.H.); nobuaki.m@aichi-med-u.ac.jp (N.M.); nobuhiro0204@hotmail.com (N.A.); 2Department of Infection, Prevention and Control, Aichi Medical University Hospital, 1-1 Yazakokarimata, Nagakute-shi 480-1195, Aichi, Japan; saka74d@aichi-med-u.ac.jp (D.S.); shibata.yuuichi.414@mail.aichi-med-u.ac.jp (Y.S.); 3Division of Biostatistics, Clinical Research Center, Aichi Medical University, 1-1 Yazakokarimata, Nagakute-shi 480-1195, Aichi, Japan; wohashi@aichi-med-u.ac.jp; 4Department of Pharmacy, Mie University Hospital, 2-174 Edobashi, Tsu-shi 514-8507, Mie, Japan; hkato59@med.mie-u.ac.jp; 5Department of Molecular Epidemiology and Biomedical Sciences, Aichi Medical University Hospital, 1-1 Yazakokarimata, Nagakute-shi 480-1195, Aichi, Japan; hagimao@aichi-med-u.ac.jp

**Keywords:** COVID-19, immunocompromised, remdesivir, monoclonal antibodies, combination

## Abstract

The coronavirus disease (COVID-19) pandemic continues to threaten global public health. Remdesivir and monoclonal antibodies have shown promise for COVID-19 treatment of patients who are immunocompromised, including those with cancer, transplant recipients, and those with autoimmune disorder. However, the effectiveness and safety of this combination therapy for patients who are immunosuppressed remain unclear. We compared the efficacy and safety of combination therapy and remdesivir monotherapy for patients with mild-to-moderate COVID-19 who were immunosuppressed. Eighty-six patients treated in July 2021–March 2023 were analyzed. The combination therapy group (CTG) showed a statistically significant reduction in viral load compared with the monotherapy group (MTG) (*p* < 0.01). Patients in the CTG also experienced earlier resolution of fever than those in the MTG (*p* = 0.02), although this difference was not significant in the multivariate analysis (*p* = 0.21). Additionally, the CTG had significantly higher discharge rates on days 7, 14, and 28 than the MTG (*p* < 0.01, *p* < 0.01, and *p* = 0.04, respectively). No serious adverse events were observed with combination therapy. These findings suggest that combination therapy may improve the clinical outcomes of immunosuppressed COVID-19 patients by reducing the viral load and hastening recovery. Further studies are required to fully understand the benefits of this combination therapy for immunocompromised COVID-19 patients.

## 1. Introduction

In late 2019, a novel coronavirus was identified as the cause of a cluster of pneumonia cases in Wuhan, a city in China’s Hubei province [1]. It spread rapidly, causing an epidemic throughout China, followed by an increasing number of cases in other countries around the world. In February 2020, the World Health Organization named the disease COVID-19, which stands for coronavirus disease. The virus that causes COVID-19 is referred to as severe acute respiratory syndrome coronavirus 2 (SARS-CoV-2). Among patients with symptomatic COVID-19, cough, myalgia, and headaches are the most commonly reported symptoms. Other manifestations, such as diarrhea, sore throat, and abnormal smell or taste are also well described [2]. The ongoing COVID-19 pandemic, caused by SARS-CoV-2, continues to pose a significant threat to public health worldwide [3]. As of 30 August 2023, the World Health Organization had reported over 770 million confirmed cases of COVID-19 and over 6.9 million deaths globally, with the numbers continuing to rise. While several treatment options for COVID-19 have been developed, including antiviral drugs and neutralizing antibodies [4,5], immunosuppressed patients, such as those with cancer, transplant recipients, and those with autoimmune disorders, including rheumatoid arthritis and Sjogren syndrome, remain at a greater risk of severe disease [6,7,8,9]. Immunocompromised patients had higher odds of intensive care unit (ICU) admission (adjusted odds ratio [aOR] = 1.40) and in-hospital death (aOR = 1.87) compared with non-immunocompromised patients [10]. Particularly, the odds of in-hospital death were higher for those with solid-organ transplants (aOR = 2.12), immunosuppressive therapy use (aOR = 1.65), or multiple myeloma (aOR = 5.28) [10]. Recent studies have reported a fatality rate reaching 8% among patients with hematological malignancy, even in cases of the Omicron variant [11,12]. Therefore, they are at a higher risk of hospitalization or death due to COVID-19 than the general population [13,14]; early treatment and close management are required [15].

Remdesivir, a ribonucleotide analog inhibitor of viral RNA polymerase, and monoclonal antibodies (mAbs) have emerged as promising treatment options for COVID-19 [16]. Clinical trials (the PINETREE study) have demonstrated that early remdesivir therapy reduces the risk of COVID-19-related hospitalizations or all-cause mortality by 87% when compared with placebo treatment on day 28 for high-risk, non-hospitalized patients [17]. mAbs treatment has also been shown to effectively reduce viral load and decrease disease progression, especially when administered early in the course of the disease [18,19]. In addition, new antiviral drugs, such as nirmatrelvir/ritonavir and molnupiravir, have recently become available for COVID-19 treatment of patients at risk of progression to severe disease, regardless of vaccination history [20].

The combination of antiviral agents with anti-spike mAbs has the potential advantage of higher efficacy owing to their different antiviral mechanisms. Some case reports and case series have reported favorable outcomes of combination therapy for patients who are immunocompromised [21,22,23]. However, the effectiveness and safety of combination therapy with remdesivir and mAbs for patients with mild-to-moderate COVID-19 who are immunosuppressed remain unclear. It is also unknown whether this combination therapy can provide more significant benefits than standard treatments for patients who are immunocompromised.

This study aimed to evaluate the clinical effectiveness and safety of combination therapy with remdesivir and mAbs compared with remdesivir monotherapy for patients with mild-to-moderate COVID-19 who were immunosuppressed. By evaluating the effectiveness of combination therapy with remdesivir and mAbs, our study has the potential to improve clinical outcomes and to reduce the burden of COVID-19 in this vulnerable population.

## 2. Materials and Methods

This retrospective, observational cohort study was conducted at Aichi Medical University Hospital (an acute care hospital with approximately 900 inpatient beds) in Aichi Prefecture, Japan. Patients aged >17 years with COVID-19 who were prescribed remdesivir alone or a combination of remdesivir and mAbs between July 2021 and March 2023 were screened. Patients requiring oxygen were excluded because mAb treatment is not recommended for patients with COVID-19 who require supplemental oxygen. Therefore, we included only patients with mild-to-moderate COVID-19 (the disease severity was evaluated in accordance with COVID-19 treatment guidelines [24]). Treatment for each group was initiated within seven days of symptom onset. The following clinical characteristics were reviewed from the medical records: age, sex, underlying disease, laboratory data, immunosuppressive status, vaccination status, clinical course, and outcomes. The definition of a patient with immunosuppression in this study was as follows: autoimmune disease with immunosuppressive drugs, kidney transplantation with immunosuppressive agents, active blood cancer, or solid tumors. Remdesivir and mAbs were prescribed if the patient had no contraindications to the drugs.

mAbs were administered on the first day of treatment, together with the first remdesivir infusion. Remdesivir was administered for 3, 5, or 10 days, depending on the attending physician’s consideration. All included patients received 200 mg of remdesivir intravenously, diluted in 250 mL of isotonic saline solution on day 1, and remdesivir 100 mg intravenously diluted in 100 mL of isotonic saline solution from day 2 to day 10 [24]. We administered remdesivir even in cases of severe renal failure (creatinine clearance of <30 mL/min), considering the low nephrotoxic potential of intravenous cyclodextrin [25]. The available mAbs administered with remdesivir were casirivimab/imdevimab and sotrovimab. Casirivimab/imdevimab was prescribed during the Delta epidemic season, while sotrovimab was prescribed during the Omicron BA1 epidemic season. Casirivimab/imdevimab was administered intravenously as a single dose of 600 + 600 mg, whereas sotrovimab was administered intravenously as a single dose of 500 mg.

The diagnosis of COVID-19 and threshold cycle (Ct) values were confirmed using real-time reverse-transcription polymerase chain reaction (RT-PCR) testing of nasopharyngeal swabs using the Cobas^®^8800 System/cobas SARS-CoV-2 (Roche Diagnostics, Basel, Switzerland) or GeneXpert^®^ System/Xpert Xpress SARS-CoV-2 (Cepheid, Beckman Coulter, Sunnyvale, CA, USA). RT-PCR testing of nasopharyngeal swabs was repeated only once within 3–7 days after treatment initiation. Patients were excluded if they had a Ct value of ≥30 at diagnosis. The variation in Ct values (a proxy for viral load) was calculated to evaluate the reduction in viral load. We statistically compared the time to an afebrile state of <37 °C; after treatment initiation and the variation in Ct values between the two groups. Possible adverse drug reactions, such as liver and kidney dysfunction related to remdesivir or mAb administration, were recorded. The prevalence of the following unfavorable outcomes was also recorded: COVID-19 exacerbation (defined as requiring oxygen supplementation after initiating therapy), COVID-19-related deaths, and 30-day all-cause mortality. This study was approved by the Human and Animal Ethics Review Committee of the Aichi Medical University Hospital, Nagakute, Japan (approval number 2022-087). The requirement for written informed consent was waived by the ethics committee owing to the retrospective nature of the study.

### Statistical Analyses

Discrete variables, such as age, were expressed as medians and interquartile ranges (IQRs). Mann–Whitney U and chi-square or Fisher’s exact tests were used to compare continuous and categorical variables, respectively. The Kaplan–Meier curve and log-rank test were performed to describe hospital discharge after 7, 14, and 28 days between combination therapy and remdesivir monotherapy.

The significance level was set at 0.05. Multivariate analysis was performed to identify factors associated with reducing fever and those associated with reducing viral load. The variables (combination therapy, being under steroid treatment, having blood cancer, and not being vaccinated against SARS-CoV-2) were selected based on their clinical relevance. Statistical analyses were performed using Statistical Package for the Social Sciences version 26 for Windows (SPSS Inc., Chicago, IL, USA).

## 3. Results

During the study period, 86 patients who met the inclusion criteria were enrolled; of these, 35 received combination therapy and 51 received remdesivir monotherapy. In the combination therapy group (CTG), 20 (57.1%) and 15 (42.9%) patients received casirivimab/imdevimab or sotrovimab, respectively. Table 1 shows a comparison of baseline characteristics between the two groups.

There were no significant intergroup differences in age, sex, or body mass index. In the groups, 3 of 35 and 1 of 51 patients had no fever during treatment. The pre-treatment body temperatures and Ct values on admission did not differ between the two groups. No significant difference was observed between the two groups with regard to underlying diseases, except that the monotherapy group was more likely to have diabetes mellitus (*p* = 0.02). Regarding laboratory data, patients in the monotherapy group (MTG) had significantly lower albumin levels than those in the CTG (*p* = 0.03). Regarding immunosuppressive status, patients in the CTG were more likely to take an immune suppressor (*p* = 0.01). In addition, patients in the CTG were more likely to have blood cancer (*p* < 0.01), while those in the MTG were more likely to have solid tumors (*p* < 0.01). Among patients with a known history of COVID-19 vaccination, there was no statistically significant difference between the two groups in terms of non-vaccination history and vaccination frequency. The mean number of days from COVID-19 onset to treatment was approximately 1.5 days in each group. There was no significant difference in the duration of remdesivir administration between the two groups.

In the univariate analysis, there were significant reductions in the mean time to resolution of fever, resolution of fever within 48 h, and resolution of fever within 72 h in the CTG (*p* = 0.02, *p* = 0.03, and *p* < 0.01, respectively). In addition, patients in the CTG showed a higher Ct value (meaning a reduction in viral load) of >3 or 5 than those in the MTG (*p* < 0.01) (Table 2). Regarding adverse events, liver dysfunction occurred in one patient in each group, kidney dysfunction occurred in one in the CTG, and infusion reactions occurred in one in the MTG (Table 3). No adverse events were observed in either group that would have necessitated discontinuation of treatment. None of the variables differed significantly regarding the incidence of COVID-19 exacerbation (requiring oxygen), death due to COVID-19, or 30-day all-cause mortality (Table 2).

In the multivariable logistic regression analysis (Table 4), combination therapy was significantly associated with higher virus reduction (odds ratio: 4.87, 95% confidence interval: 1.629–14.555, *p* < 0.01) but was unrelated to afebrile status within 72 h.

No significant difference was observed between the patients treated with sotrovimab and those treated with casirivimab/imdevimab in terms of fever resolution or viral load reduction (data are not shown).

We also compared the length of hospital stays between the two cohorts to analyze the effect of combination therapy. Of the 86 patients, we excluded 16 patients who were infected during hospitalization for other reasons. In addition, we excluded eight patients who died during hospitalization because we were unable to collect their discharge data. In total, we included 32 and 30 patients in the CTG and MTG, respectively. Patients treated with combination therapy had a shorter median hospital stay compared with those treated with monotherapy, although no significant differences were found (7.0 [IQR 5.0–11.0] vs. 11 [9,10,11,12,13,14,15], *p* = 0.31). However, patients treated with combination therapy had a significantly higher discharge rate on days 7, 14, and 28 than those treated with monotherapy (Figure 1A–C).

## 4. Discussion

In this study, we evaluated the efficacy and safety of combination therapy (remdesivir and mAbs) compared with those of remdesivir monotherapy for patients with mild-to-moderate COVID-19 who were immunosuppressed. A notable aspect of our research was that we found a statistically significant increase in the Ct value (indicating a reduction in viral load) among patients in the CTG compared with those in the MTG. In addition, only patients in the MTG required oxygen supplementation and experienced COVID-19-related death, although there was no statistically significant difference between the two groups. Moreover, our study findings add to the limited data available on the safety of combination therapy for patients with COVID-19, as we found no significant adverse events associated with its use. To date, combination therapy has only been reported in single cases and small series [21,22,23]. To the best of our knowledge, this is the first study to examine the medical outcomes and safety of combination therapy with remdesivir and mAbs compared with those of monotherapy with remdesivir against COVID-19 for patients who are immunocompromised. Our results suggest that combination therapy has the potential to improve clinical outcomes and reduce the burden of COVID-19 in this vulnerable population.

Recent systematic reviews regarding the immunological efficacy of COVID-19 vaccines in patients with immunosuppressive status, such as those with cancer or hematological malignancy, organ transplant recipients, and those on corticosteroids, showed an impaired response to SARS-CoV-2 vaccination [26], and immunocompromised individuals are known to have an increased rate of severe and fatal outcomes related to COVID-19 [27]. To maximize the immune response, these populations should receive more vaccine doses than the general population; the CDC recommends that individuals with immunocompromising conditions should receive at least three vaccine doses [28]. In our study population, the CTG patients who were vaccinated less than twice accounted for 43% of the participants, meaning these patients were negative or had lower anti-SARS-CoV-2 serology. However, no patient experienced COVID-19 exacerbation or death due to COVID-19 in the CTG, although there was no significant difference compared to the MTG. Therefore, for immunocompromised patients with low vaccination frequency, combination therapy could be considered one treatment option. Riccardo et al. investigated remdesivir alone or in combination with mAbs as an early treatment for preventing severe COVID-19 in patients with mild or moderate disease among high-risk populations, such as those with obesity, immunodeficiency, negative SARS-CoV-2 serology, and lymphocyte counts of ≤1000 cell/μL [29]. In their single-center study, they compared 30 patients who were treated with remdesivir to 32 patients who received remdesivir plus mAbs. Among them, the former included 18 (60%) patients who were immunosuppressed, and the latter included 18 (56.3%) who were immunocompromised. They found no difference in the rate of disease progression (hospitalization, increase in oxygen supplementation, ICU admission, and death) between the two groups, although patients in the combination group were more likely to have negative anti-SARS-CoV-2 serology (37% versus 10%, *p* = 0.04) and lower lymphocyte counts (*p* = 0.02) than those treated with remdesivir alone. In the current study, no significant difference in outcomes (oxygen requirement, COVID-19-related death, and 30-day all-cause mortality) was observed between the two groups, despite the fact that the patients in the CTG experienced more hematological cancer complications and received more immunosuppressive medications than those in the MTG. Therefore, we believe that combination therapy contributes particularly to hematological populations and populations taking immunosuppressive drugs, although larger studies are needed to assess the efficacy and safety of combination therapy in immunocompromised populations.

In this study, univariate analysis showed a statistically significant resolution of fever in the CTG compared with the MTG, although multivariate analysis did not. The small sample size of the CTG may have affected the results of the multivariate analysis. Another suspected factor may be that the duration of remdesivir administration differed between the two groups, although this difference was not statistically significant. Therefore, future research should be conducted with a larger sample size and the same duration of remdesivir treatment between the two groups.

Immunocompromised patients are more likely to develop persistent and prolonged COVID-19 because of their weak immune systems, compared to the immunocompetent population [30]. Previous studies have revealed that patients who are immunosuppressed have a prolonged viral shedding period and may be at a higher risk of infections by the new SARS-CoV-2 variants [31,32,33]. Lee et al. also characterized the SARS-CoV-2 evolution in patients who were immunosuppressed and experienced long-term SARS-CoV-2 shedding and found that candidate variants were likely to eventually spread to the population [34]. In addition, persistent SARS-CoV-2 infection of these individuals can lead to drug resistance and immunological escape [35]. Therefore, it is particularly important to promptly reduce the viral load of patients who are immunosuppressed, and it was found that combination therapy resulted in an early increase in the Ct value (meaning a reduction in viral load) compared with remdesivir monotherapy. In addition to researching the efficacy of combination therapy for reducing the duration of long COVID-19, the duration of infectious viral shedding in immunosuppressed patients also needs to be studied in the future, with serial sampling to determine how much shorter the duration of infectious viral shedding and prevention of infections caused by the variants are with combination therapy compared with monotherapy.

We assume that the underlying mechanism of reducing the viral load earlier in the CTG is that remdesivir and mAbs interfere with two different stages in the replication cycle of SARS-CoV-2. The antiviral drug, remdesivir, interferes with RNA-dependent RNA polymerase, thereby blocking SARS-CoV-2 RNA replication [36]. mAbs bind to the viral spike glycoprotein, blocking viral attachment and entry into host cells [37,38]. Thus, combining antiviral agents with mAbs can have a potential advantage of higher efficacy due to their different antiviral mechanisms, as inhibition of viral proliferation might be insufficient for viral clearance in the absence of humoral immunity [39,40]. Data on the treatment of patients who are immunosuppressed with this combination of drugs are lacking. Nicola et al. retrospectively evaluated the efficacy and safety of a combination of casirivimab/imdevimab and remdesivir for the treatment of severe COVID-19 in a small population (all patients tested negative for anti-SARS-CoV-2 spike protein antibodies) [41]. They found that all but 1 of the 14 patients experienced an improvement in clinical status and respiratory parameters within 7 days of treatment, with no adverse drug reactions [41], and emphasized that the mortality rate among patients in the combination therapy cohort was favorable compared with that of patients with a similar comorbidity profile. In addition, Magyari et al. found that B-cell-depleted patients (20 patients who demonstrated undetectable baseline anti-SARS-CoV-2 immunoglobulin levels before treatment) with COVID-19 pneumonia who were treated with remdesivir and anti-SARS-CoV-2 immunoglobulins simultaneously required a significantly shorter time for PCR positivity, oxygen weaning, and length of hospital stay compared to patients who received remdesivir and anti-SARS-CoV-2 immunoglobulins consecutively [42]. Dioverti et al. also reported that a combination of mAbs and remdesivir was effective for and well tolerated by B-cell-depleted patients [23]. Shimizu et al. reported that a patient under rituximab therapy had a successful outcome after combination therapy, although multiple treatments with remdesivir failed to achieve a cure [21]. Moreover, although the RECOVERY study was not designed to evaluate the benefits of combination therapy with casirivimab/imdevimab and remdesivir, it found the less frequent progression to ventilator use and reduced mortality among seronegative patients treated with this combination [43]. Taken together, our study and previous studies [41,42,43] have demonstrated that the concept of combining remdesivir with anti-SARS-CoV-2 mAbs for patients who are immunosuppressed, who are likely to have an impaired response to SARS-CoV-2 vaccination, is valid because it could simultaneously lead to a synergistic interaction with an increase in therapeutic efficacy not only in patients with mild-to-moderate COVID-19 but also in those with severe COVID-19.

Regarding the length of hospitalization, the estimated rates of hospital discharge at days 7, 14, and 28 were higher in the CTG than in the MTG, which is consistent with the study by Andrea et al. [44]. In their study, among 314 COVID-19 patients treated with sotrovimab (33.1% had immune depression, 7% were transplant recipients, and 27.3% had oncological disease), 96 patients (30.5%) were administered antiviral drugs in addition to sotrovimab. They found that patients treated for COVID-19 had a lower median hospital stay and higher discharge rate on days 7 and 14 compared with those not treated.

This study has several strengths. First, this study focused on a specific population (patients with mild-to-moderate COVID-19 who were immunosuppressed) that has not been extensively studied in previous COVID-19 clinical trials, and it is the first to examine real-world medical outcomes and the safety of combination therapy with remdesivir and mAbs compared with those of monotherapy with remdesivir. It is notable that the study found a significant increase in the Ct values (indicating a reduction in viral load) in the CTG compared with the MTG. By providing evidence of the effectiveness and safety of combination therapy in this patient population, this study has the potential to inform clinical practice and improve outcomes for those who are immunosuppressed and are at the highest risk of developing severe COVID-19. The results of this study may have implications for the management of COVID-19 in vulnerable populations, and smaller studies can provide valuable preliminary data that can inform more extensive studies in the future. In addition, the results of this study can be considered a starting point for further investigation of the efficacy of combination therapy with remdesivir and mAbs against COVID-19 in patients with immunosuppression.

Our study has a few limitations. First, the generalizability of the findings may be limited, as this study was retrospective, had a small sample size, and was conducted at a single center. However, randomized controlled trials with larger sample sizes may not be feasible in specific clinical settings or patient populations, such as immunocompromised hosts. Second, although each mAb was administered when it was active against prevalent viruses, no sequencing was performed to determine the viral variants after obtaining the nasopharyngeal swabs. Due to the changing SARS-CoV-2 variant strains, it is unclear whether results similar to those of this study can be obtained. After studying the SARS-CoV-2 strains, further research should be conducted on combination therapy using mAbs that are effective against prevalent SARS-CoV-2 strains, including mutant strains. Third, viral cultures were not performed, and we used Ct values as a proxy for viral load because viral cultures are not routinely conducted in most laboratories. Several previous studies have suggested that the Ct values correlate with the viral load and could be indicators of viral load [45,46]. Lower Ct values are associated with worse outcomes and are helpful for predicting the clinical course and prognosis of patients with COVID-19 [46]. Therefore, we believe this study, which investigated the effects of combination therapy by using Ct values, is useful. Fourth, we did not evaluate serum SARS-CoV-2 immunoglobulin G levels in the cohort patients. The administration of mAbs may be less effective for patients with antibodies against SARS-CoV-2. Therefore, additional studies on combination therapy with remdesivir and mAbs should be conducted in the SARS-CoV-2 seronegative population in the future.

## 5. Conclusions

Our study adds to the growing body of evidence on the efficacy of combination therapy with remdesivir and mAbs, which may be a promising approach for the treatment of COVID-19 in immunosuppressed patients. The significant increase in the Ct value (indicating a reduction in viral load) observed with combination therapy suggests that this approach may lead to more rapid clinical improvement. The results of this study suggest that combination therapy could be one of the safe treatment options for COVID-19 in immunocompromised patients. In addition, physicians may consider administering combination therapy in cases of high viral load, as it results in an early reduction in viral load. However, further studies are required to fully understand the benefits of this combination therapy.

## Figures and Tables

**Figure 1 viruses-15-01952-f001:**
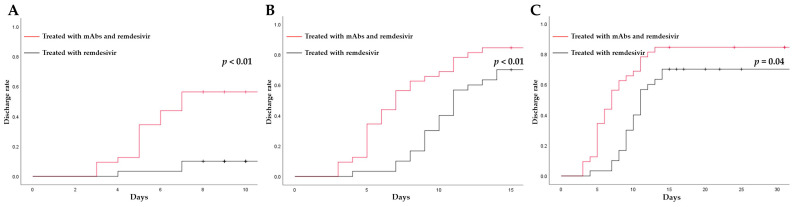
(**A**) The 7-day hospital discharges of CTG and MTG patients. (**B**) The 14-day hospital discharges of CTG and MTG patients. (**C**) The 28-day hospital discharges of CTG and MTG patients. CTG, combination therapy group; mAbs, monoclonal antibodies; MTG, monotherapy group.

**Table 1 viruses-15-01952-t001:** Comparison of baseline characteristics between the two groups.

Variables	All (*n* = 86)	Combination (*n* = 35)	Remdesivir Alone (*n* = 51)	*p*-Value
Age (years), median (IQR)	70.5 (55–79)	66 (46–78)	74 (63–80)	0.12
Female sex (*n*, %)	35 (40.7)	14 (40.0)	21 (41.2)	0.91
Body mass index, median (IQR)	20.6 (18.9–22.9)	20 (18.1–22.7)	21.5 (19.3–23.0)	0.21
Body temperature, median (IQR), *n* = 82	38.2 (37.6–38.7)	38.0 (37.3–38.9)	38.4 (37.7–38.6)	0.34
Ct value on admission	19.8 ± 4.5	19.5 ± 4.6	20.8 ± 5.0	0.23
Underlying disease:				
-Cardiovascular disease	6 (7.0)	1 (2.8)	5 (9.8)	0.21
-Asthma	3 (3.5)	1 (2.8)	2 (3.9)	0.79
-Chronic obstructive pulmonary disease	5 (5.8)	3 (8.6)	2 (3.9)	0.36
-Diabetes mellitus	14 (16.3)	2 (5.7)	12 (23.5)	0.02
-Hypertension	28 (32.6)	9 (25.7)	19 (37.2)	0.26
-Hyperlipidemia	8 (9.3)	1 (2.8)	7 (13.7)	0.08
-Chronic kidney disease	14 (16.3)	6 (17.1)	8 (15.7)	0.85
-Hemodialysis	2 (2.3)	1 (2.8)	1 (2.0)	0.78
Laboratory data, median (IQR):				
-White blood cells (μL)	6200 (4100–8400)	4900 (3850–7900)	6700 (4450–9050)	0.19
-Neutro	4825 (3093–6580)	4081 (3093–6509)	5040 (3137–6935)	0.27
-Lympho	867 (483–1258)	597 (418–928)	941 (690–1324)	0.09
-CRP	2.4 (0.8–5.3)	1.5 (0.5–3.5)	3.3 (1.1–6.9)	0.06
-T-Bil	0.6 (0.4–0.8)	0.6 (0.4–0.9)	0.6 (0.4–0.8)	0.71
-AST	24 (20–43)	21 (17–29)	29 (21–49)	0.11
-ALT	19 (12–35)	15 (12–22)	22 (14–38)	0.18
-LDH	217 (188–285)	206 (184–269)	222 (200–324)	0.56
-Creatinine	0.8 (0.6–1.1)	0.8 (0.7–1.0)	0.8 (0.6–1.0)	0.97
-Albumin	3.4 (2.8–3.9)	3.7 (3.1–4.0)	3.2 (2.7–3.7)	0.03
-D-dimer	1.5 (1.0–3.5)	1.1 (0.9–1.6)	2.5 (1.2–4.8)	0.11
Autoimmune disease	17 (19.8)	6 (17.1)	11 (21.5)	0.61
Immune suppressor:	38 (44.1)	21 (60)	17 (26.1)	0.01
-Steroids	29 (33.7)	16 (45.7)	13 (25.5)	0.05
-Ciclosporin	7 (8.1)	5 (14.3)	2 (3.9)	0.08
-Azathioprine	2 (2.3)	2 (5.7)	0	0.08
-Methotrexate	4 (4.6)	2 (5.7)	2 (3.9)	0.69
-Mycophenolate mofetil	9 (10.5)	6 (17.1)	3 (5.9)	0.09
-Iguratimod	1 (1.2)	1 (2.8)	0	0.22
-Belimumab	2 (2.3)	2 (5.7)	0	0.08
-Pomalidomide	1 (1.2)	1 (2.8)	0	0.22
-Salazosulfapyridine	1 (1.2)	1 (2.8)	0	0.22
-Tacrolimus hydrate	5 (5.8)	2 (5.7)	3 (5.9)	0.97
-Mesalazine	1 (1.2)	0	1 (1.9)	0.40
-Everolimus	1 (1.2)	0	1 (1.9)	0.40
-During chemotherapy	8 (9.3)	5 (14.3)	3 (5.9)	0.18
Kidney transplantation	9 (10.5)	6 (17.1)	3 (5.9)	0.09
Blood cancer	19 (22.1)	13 (37.1)	6 (11.7)	<0.01
Solid tumor	38 (44.2)	9 (25.7)	29 (56.8)	<0.01
-Stage IV	22 (25.6)	6 (17.1)	16 (31.4)	0.13
Vaccination for COVID-19:				
-Non-vaccinated	17 (19.7)	4 (11.4)	13 (25.4)	0.10
-One time	2 (2.3)	2 (5.7)	0	0.08
-Two times	15 (17.4)	9 (25.7)	6 (11.7)	0.09
-More than three times	46 (53.4)	19 (54.2)	27 (52.9)	0.90
-Unknown	6 (6.9)	1 (2.8)	5 (9.8)	0.21
Days from onset to start of treatment, median (IQR)	1.0 (1.0–2.0)	1.0 (1.0–2.0)	1.0 (0–2.0)	0.79
Duration of remdesivir administration:				
-3 days	53 (61.6)	18 (51.4)	35 (68.6)	0.10
-5 days	25 (29.1)	14 (40)	11 (21.5)	0.06
-10 days	8 (9.3)	3 (8.6)	5 (9.8)	0.84

ALT, Alanine transaminase; AST, Aspartate transaminase; COVID-19, coronavirus disease 2019; CRP, C-reactive protein; Ct, threshold cycle; IQR, interquartile range; LDH, lactate dehydrogenase; T-bil, total bilirubin.

**Table 2 viruses-15-01952-t002:** Comparison of clinical efficacy and outcomes between the two groups.

Variables	All (*n* = 86)	Combination (*n* = 35)	Remdesivir Alone (*n* = 51)	*p*-Value
Clinical efficacy of treatment:				
-Time for fever resolution of <37 °C of at least 24 h, median (IQR), *n* = 82	37.0 (21.0–76.1)	27.5 (21.2–60.2)	56.5 (20.5–90.7)	0.02
-Afebrile within 48 h, *n* = 82	43 (52.4)	20 (71.4)	23 (46)	0.03
-Afebrile within 72 h, *n* = 82	57 (69.5)	26 (92.8)	31 (62)	<0.01
-Increase in Ct value of >3 after initiating treatment	53 (61.6)	28 (80)	25 (49)	<0.01
-Increase in Ct value of >5 after initiating treatment	48 (55.8)	27 (77.1)	21 (41.1)	<0.01
Outcomes:				
-COVID-19 exacerbation (requiring oxygen after initiating treatment)	3 (3.5)	0	3 (5.9)	0.22
-Death by COVID-19	2 (2.3)	0	2 (3.9)	0.32
-30-day all-cause mortality	8 (9.3)	2 (5.7)	6 (11.7)	0.34

COVID-19, coronavirus disease; IQR, interquartile range.

**Table 3 viruses-15-01952-t003:** Comparison of adverse events between the two groups.

Variables	All (*n* = 86)	Combination (*n* = 35)	Remdesivir Alone (*n* = 51)	*p*-Value
Adverse event:	4 (4.6)	2 (5.7)	2 (3.9)	0.69
-Liver dysfunction	2 (2.3)	1 (2.8)	1 (1.9)	0.78
-Kidney damage	1 (1.2)	1 (2.8)	0	0.22
-Infusion reaction	1 (1.2)	0	1 (1.9)	0.40

**Table 4 viruses-15-01952-t004:** Multivariable logistic regression analysis between the two groups.

Afebrile within 72 h			
Variables	Odds Ratios	95% Confidence Interval	*p*-Value
-Combination therapy	2.085	0.657–6.609	0.21
-Steroid	1.02	0.342–3.04	0.97
-Blood cancer	1.764	0.417–7.467	0.44
-Non-vaccinated	0.764	0.235–2.486	0.65
**Increase in Ct value of >5**			
Variables	Odds Ratios	95% Confidence Interval	*p*-Value
-Combination therapy	4.87	1.629–14.555	<0.01
-Steroid	2.231	0.773–6.443	0.13
-Blood cancer	0.748	0.213–2.63	0.65
-Non-vaccinated	1.216	0.376–3.931	0.74

## Data Availability

The datasets used and/or analyzed in the current study are available from the corresponding author on reasonable request.

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
