# Peer review of "Real-World Experience of the Comparative Effectiveness and Safety of Combination Therapy with Remdesivir and Monoclonal Antibodies versus Remdesivir Alone for Patients with Mild-to-Moderate COVID-19 and Immunosuppression: A Retrospective Single-Center Study in Aichi, Japan"

_viruses, 2023, doi:10.3390/v15091952_

Round 1

Reviewer 1 Report

1.           Before discussing treatments, it's important to provide a background on what COVID-19 is. I recommend detailing both the major symptoms (fever, dyspnea, cough) and the minor symptoms (dysgeusia, anosmia, gastrointestinal symptoms, headache, and skin lesions). This will offer the reader a complete understanding of the disease. For additional information, you could refer to the following articles: DOI: 10.1371/journal.pone.0248009, DOI: 10.1097/IPC.0000000000000952, DOI: 10.1002/hed.26269.

2.           Line 37-42: Your current phrasing can be improved for clarity. For example: While several treatment options for COVID-19 have been developed—including antiviral drugs and neutralizing antibodies—immunosuppressed patients remain at a greater risk. This group includes individuals with cancer, transplant recipients, and those with autoimmune disorders such as rheumatoid arthritis and Sjogren syndrome. They are more likely to develop severe COVID-19.

3.           Line 46: You have omitted Nirmatrelvir/ritonavir and Molnupiravir. Consider adding a sentence about these antiviral drugs. For references, you can use: DOI: 10.3390/v15010071, DOI: 10.1056/NEJMoa2116044, DOI: 10.1002/jmv.28011, DOI: 10.1056/NEJMoa2204919.

4.           Line 102: Clarify the timing for evaluating the increase in Ct values. When were the subsequent swabs taken?

5.           Line 117: Please remove “, and p < 0.05 was significant.”

6.           Statistical Methods: For continuous variables, use either the median or mean according to data distribution. Given the small sample size, it is likely that your data for age is not normally distributed. In such cases, report only the median (IQR) and use non-parametric tests for comparison, like the Mann-Whitney U test. The same applies to BMI, body temperature, Ct values, and blood tests.

7.           Table 1:

-             The current table is lengthy and hard to read. Consider simplifying it by only including categories of diseases (e.g., Autoimmune disease, blood cancer, etc.) and moving details to supplemental material.

-             Create a table 2 for Clinical efficacy of treatment, and outcomes, and a table 3 for the adverse events.

-             It is not clear the timing for the increasing value of Ct.

8.           I understand that the limited sample size precludes a robust multivariate logistic analysis for assessing disease progression, necessity of starting oxygen therapy, or death as outcomes. That said, the focus on patients becoming afebrile within 72 hours may not offer substantial insights.

Instead, have you considered evaluating the length of hospitalization as a relevant outcome measure? If you have the data, conducting a Cox hazard analysis and generating a Kaplan-Meier survival curve could provide valuable insights into treatment efficacy and patient outcomes. I believe this approach could significantly enhance the overall impact of your paper.

9.           Have you seen difference between sotrovimab and casirivimab/imdevimab?

10.         I suggest you to read this recent paper that perform a similar analysis on sotrovimab (10.3390/v15081757) and add it in the discussion.

The manuscript is comprehensive. However, your it could benefit from improved structure and formatting for easier comprehension.

Reviewer 2 Report

Authors written an interesting paper on COVID-19 therapies. The data are well presented and the paper is clear and metodholocically correct . In addiction the topic is relevant and well described.

Below my minor suggestions:

1. INtroduction: add data from global burden of COVID-19 cases at the day of resubmission and which population are at risk of diseases progressions 

2. Methods and results clear

3. Discussion: discuss better the role protective of vaccination. Furthermore, discuss the differnt action of Ab monoclonal and RDV especially useful in hematological and fragile patients. Ando also heamtochimal risk factors such as anemia (Anemia as a risk factor for disease progression in patients admitted for COVID-19: data from a large, multicenter cohort study. Sci Rep. 2023 Jun 3;13(1):9035. doi: 10.1038/s41598-023-36208-y. )

Please add limitation section where you recognize limitation of the study such as retrospective, mono centric study...

In discussion also consider the impact on long covid consequences (see Incidence of long COVID-19 in people with previous SARS-Cov2 infection: a systematic review and meta-analysis of 120,970 patients. Intern Emerg Med. 2023 Aug;18(5):1573-1581.)

Conclusion: give some infectious diseases therapeutic  proposal that came from your interesting and well written paper. 

Please minor English revision is needed
